# Nuclear-cytoplasmic Shuttling in Chronic Myeloid Leukemia: Implications in Leukemia Maintenance and Therapy

**DOI:** 10.3390/cells8101248

**Published:** 2019-10-14

**Authors:** Giovanna Carrà, Isabella Russo, Angelo Guerrasio, Alessandro Morotti

**Affiliations:** Department of Clinical and Biological Sciences, University of Turin, Regione Gonzole 10, 10043 Orbassano (Turin), Italy; isabella.russo@unito.it (I.R.); angelo.guerrasio@unito.it (A.G.)

**Keywords:** nuclear-cytoplasmic shuttling, chronic myeloid leukemia, BCR-ABL, miRNA, PTEN, p53

## Abstract

Nuclear-cytoplasmic shuttling is a highly regulated and complex process, which involves both proteins and nucleic acids. Changes in cellular compartmentalization of various proteins, including oncogenes and tumor suppressors, affect cellular behavior, promoting or inhibiting proliferation, apoptosis and sensitivity to therapies. In this review, we will recapitulate the role of various shuttling components in Chronic Myeloid Leukemia and we will provide insights on the potential role of shuttling proteins as therapeutic targets.

## 1. Introduction

Shuttling from the nucleus to the cytoplasm and back is a highly regulated and complex process, which may include proteins and nucleic acids such as mRNA molecules [1,2,3]. This process has been linked to the regulation of cell cycle induction and cellular proliferation with several implications in both normal and pathological cellular growth. Nuclear-cytoplasmic shuttling is a targetable process, with potentially relevant implications from the therapeutic standpoint [4,5]. Very recently, through a fast track designation, the Food and Drug Administration (FDA) approved the exportin 1 (XPO1) inhibitor selinexor in multiple myeloma patients with a refractory disease to at least one proteasome inhibitor, one immunomodulatory agent, and daratumumab (triple-class refractory) [6]. In such a demanding scenario, selinexor displayed promising response rates, suggesting that the nuclear-cytoplasmic shuttling represents a master target in cancer therapy. 

Several tumor suppressors and oncogenes have been recognized to affect tumorigenesis through their ability to shuttle among various cellular compartments. For instance, the tumor suppressors PTEN, p53 and FOXOs may lose their tumor suppressive functions if delocalized [7]. The identification of shuttling tumor suppressors and the mechanisms that promote their delocalization in the cell may allow the development of therapies to restore the proper cellular compartmentalization with consequent re-activation of the tumor suppressive functions [8]. In this respect, it is worth noting that the reactivation of tumor suppressor is the most efficient approach to promote cancer apoptosis, as it was pointed out for p53 [9]. As a consequence, strategies to restore the correct cellular compartmentalization are highly promising therapeutic approaches [5].

Chronic Myeloid Leukemia (CML) is a myeloproliferative disorder characterized by the translocation t(9;22), which codes for the chimeric protein BCR-ABL [10]. This disease is the paradigm of the precision medicine, due to the stunning efficacy of BCR-ABL inhibitors [11]. However, it should be noted that three major issues may limit the efficacy of TKI-based therapies. First, in rare cases, BCR-ABL develops mutations that render CML resistant to TKI treatment [12]; second, TKIs based therapies are unable to completely eradicate the disease, due to the resistance of CML stem cells to TKI [13]; third, CML blast crisis, and other Ph+ leukemias such as ALL, are barely unaffected by TKI therapy [14]. 

As a consequence, the understanding of those mechanisms that promote CML maintenance should identify novel targets and novel therapies to enforce TKI-based regiments. 

Shuttling appears to be an essential process to look at in the CML context, due to the ability of c-ABL and BCR-ABL to shuttle between the nucleus and the cytoplasm. BCR-ABL was indeed described as a cytoplasmic protein although it contains three nuclear localization sequences (NLS) [15] and a nuclear export sequence (NES) [16]. Various studies have clearly demonstrated that c-ABL, one of the partner of the chimeric BCR-ABL protein, is indeed a shuttling protein, especially upon DNA damage [17]. The presence of BCR protein in the fusion with ABL was conversely shown to promote a cytoplasmic localization, due to the binding with actin [18] and a specific conformation status with the kinase domain [19]. Several years ago, it was however shown that the inhibition of BCR-ABL by TKI was also associated with the enrichment of BCR-ABL into the nucleus [20], and that this localization was further augmented by the concomitant treatment with leptomycin B, a known nuclear export inhibitor. Authors demonstrated that only kinase-defective BCR-ABL or TKI-inhibited BCR-ABL entered the nucleus, while active full length BCR-ABL was unable [20,21]. Notably, nuclear BCR-ABL was shown to promote apoptosis in CML cellular models [20] and primary samples [22] when reactivated upon TKI removal. All together these data clearly point to the issue that the leading oncogene of CML may turn into a suicide gene, when delocalized, and that shifting cellular compartments is achievable with the combination of various drugs. 

These observations point to the relevance of studying nuclear-cytoplasmic shuttling in CML and therefore this review will investigate the nuclear-cytoplasmic scenario of proteins and RNAs molecules in CML.

## 2. Shuttling Tumor Suppressors

### 2.1. PTEN

The tumor suppressor PTEN is one of the most frequently mutated/deleted tumor suppressors in cancer. PTEN is known to play a role both in the cytoplasm, as a negative regulator of the PI3K-AKT pathway, and in the nucleus where it targets other pathways [23,24]. Shuttling from the nucleus and the cytoplasm was associated with PTEN mono-ubiquitination [25,26] or sumoylation [27]. The regulation of PTEN shuttling through mono-ubiquitination is modulated by the de-ubiquitinase USP7 [25]. We demonstrated that BCR-ABL is able to promote USP7 activation through tyrosine phosphorylation, allowing to affect PTEN cellular compartmentalization [28,29]. Notably, we observed differential PTEN cellular compartmentalization in the stem cell pool, where it is predominantly nuclear, when compared to a predominant cytoplasmic localization in the more differentiated cells [28]. This observation may contribute to explain the quiescence of the stem cell pool and provide implications from the therapeutic standpoint, as specifically reviewed [30]. 

### 2.2. Foxo 

The Forkhead box O (FOXO) is one subfamily of the fork head transcription factor family with important roles in tumorigenesis [30,31]. FOXOs proteins have been extensively studied as PI3K-Akt targets with remarkably roles in cancer [32] and indeed as potential candidates for therapies [33]. 

In CML, FOXO proteins were shown to be nuclear excluded [34,35], and to play a pivotal role in mediating the quiescence of CML stem cells. In particular, FOXO1 and FOXO3a were shown to be inactivated and delocalized into the cytoplasm by BCR-ABL. Notably, treatment with TKI reduces FOXO phosphorylation, favoring relocalization to the nucleus. As a consequence, active FOXOs can modulate the expression of Cyclin D1, ATM and other genes that promote cell cycle arrest and apoptosis induction. Various mechanisms have been described to promote FOXO shuttling between the nucleus and the cytoplasm [36] and some drugs have also been described to affect shuttling [37]. It is worth to note that changes in FOXO differentiate the CML stem cell pool from the progenitor pool, as observed for PTEN. Such variations in localization may therefore be utilized to target the stem cell fraction [13].

### 2.3. P53 

The tumor suppressor p53 is recognized as one the most important tumor suppressors in cancer [38]. While generally known as a transcriptional factor, it was also shown to play a role in the cytoplasm [39]. As a consequence, shuttling of p53 from the nucleus to the cytoplasm affects its role in tumorigenesis [7,40,41]. The tumor suppressor p53 is essential in CML pathogenesis, even if TP53 mutations were discovered only during the progression of the disease [42,43]. In a recent observation, p53 protein was clearly described as a master regulator of CML stem cells, in a tight correlation with c-Myc [44]. In line with these observations, we have shown that BCR-ABL favors p53 nuclear exclusion, causing inactivation of its tumor suppressive functions [45]. In accordance, p53 was shown to be mostly cytoplasmic in CML CD34 positive cells, with a further cytoplasmic accumulation upon imatinib and treatment with DNA-damaging agents [46]. Similarly, cytoplasmic p53 was shown to accumulate in CML progenitor cells, upon treatment with an inhibitor of SIRT1 [47]. While these works have shown p53 accumulation in the cytoplasm of CML cells, it is worth to note that others did not confirm such localization [44]. However, CML is a disease characterized by a highly complex mixture of cells at different stages of differentiation and therefore it could be speculated that the localization of p53 in the cytoplasm or in the nucleus may depend on the differentiation status of the cells, as we have observed for PTEN [28]. A further investigation is indeed required to better assess how and when p53 shuttles between the nucleus and the cytoplasm of CML cells.

### 2.4. P27

p27 is an inhibitor of the cyclin-dependent kinases and is therefore involved in the regulation of cell cycle [48]. As the majority of cell cycling inhibitors, p27 functions are tightly regulated at transcriptional and post-transductional level, as reviewed [49]. Furthermore, shuttling into the cytoplasm was shown to abrogate p27 function as inhibitor of cyclin. Mechanisms that affects stability or cellular compartmentalization of p27 may indeed affect tumorigenesis. In a recent report, BCR-ABL was shown to promote p27 shuttling from the nucleus to the cytoplasm in a kinase independent manner [50]. Notably, beside the loss of the tumor suppressive function in the nucleus, the localization in the cytoplasm affects p27 functionality promoting an oncogenic role, although the mechanism still must be identified. Some reports have suggested the ability of p27 to modulate RhoA activity [51]. 

## 3. BCR-ABL-Dependent Shuttling of RNA-Binding Proteins

The regulation of mRNA processing can rapidly and efficiently modulate the function of specific proteins, without altering gene expression [52]. Indeed, different types of cancer are characterized by the modulation of RNA-binding proteins (RBP), that are responsible of the processing, nuclear export, regulation of stability, and translation of mRNA molecules [53]. Among various processes, nuclear export of specific mRNA molecules is associated with neoplastic transformation. Not surprisingly, altered expression of some RBPs with nuclear-cytoplasmic shuttling activity has been shown in BCR/ABL leukemogenesis [54]. In particular, the expression of RNA-binding proteins regulates various cellular processes and indeed it appears clear that the deregulation of the expression of RBP may affect leukemogenesis, through the impairment of mRNA metabolism. As a consequence, several RBPs have been implicated in the pathogenesis and in the progression of CML [55,56].

## 4. Ribonuclear Proteins

### 4.1. hnRNP A1

hnRNP A1 is a ubiquitously expressed hnRNP, and is directly involved in the nucleocytoplasmic trafficking of mRNA molecules [57]. Primarily, hnRNP A1 is nuclear, but it shuttles continuously between the nucleus and the cytoplasm, where dissociates from its mRNA [58]. hnRNP A1 expression is higher in transformed cell lines compared to normal and differentiated tissues [59]. In BCR/ABL-expressing cells hnRNP A1 protein levels are markedly increased as well as in CML-BC samples compared to CML-CP samples and appears to correlate with BCR/ABL levels [60]. Mechanistically, hnRNP A1 is activated and stabilized by the PI3K and BCR/ABL-regulated PKC, with further increase in its ability to shuttle mRNAs from the nucleus. 

Interference with hnRNP A1 shuttling activity resulted in down-regulation of C/EBPalpha, the major regulator of granulocytic differentiation, Bcl-X_L_, an important survival factor for hematopoietic cells and SET, the inhibitor of protein phosphatase 2A (PP2A) [60]. 

### 4.2. Fus

FUS is a hnRNP protein also known as hnRNP P2, originally discovered as the N-terminal part of a fusion gene with CHOP in myxoid liposarcoma carrying the translocation t(12;16). FUS is primarily localized in the nucleus where it is involved in nucleocytoplasmic shuttling of specific RNA [61,62]. BCR/ABL kinase activity is essential for FUS expression and DNA-binding through the induction of the nuclear PKC−II Isoform, that prevents its degradation. Ectopic FUS expression in 32Dcl3 cells was associated with proliferation and reduced sensitivity to apoptotic stimuli. On the contrary, FUS down regulation reduces proliferation and enhances the susceptibility of BCR/ABL-expressing cells to apoptosis. These effects mostly depend on decreasing level of G-CSFR, suggesting that FUS interferes with the export of G-CSFR mRNA to the cytoplasm [63]. 

## 5. BCR/ABL-Dependent Shuttling of Non-Coding RNA 

Several recent studies provide evidences that BCR-ABL regulates the traffic of exosomes, riches in RNA and proteins, between cells and the bone marrow microenvironment. Exosome traffic is not to be considered as a classical nuclear-cytoplasmic shuttling, but it is a mechanism through which the nucleus of a cell communicates with the cytoplasm of surrounding cells [64]. Therefore, we describe it as an “atypical” form of nuclear-cytoplasmic shuttling. In this section, we include some examples of miRNA that are delivered from the nucleus of one cell to the surrounding cells. 

### 5.1. miRNA-126

It was demonstrated that the CML LAMA84 cell line releases exosomes into the medium culture [65] and that these exosomes are able to modulate gene expression of endothelial cells through the release of miRNAs by exosomes. In particular, the analysis of miRNAs expression profile showed that miR-126 was highly enriched in LAMA84 exosomes. Ectopic expression of miR-126 in LAMA84 and co-culture with Human umbilical vein endothelial cells (HUVEC) demonstrated the shuttle of miR-126 in endothelial cells, where it down-modulates two target genes, CXCL12 and VCAM1 [66]. 

### 5.2. miRNA-210

Cellular and exosomal miRNA profiling of K562 cells cultured in hypoxic conditions allowed to identify in miR-210 one of the most upregulated miRNA. Exosomal miR-210 was shown to target the gene codifying Ephrin-A3, an anti-angiogenic factor able to modulate VEGF signaling [67].

### 5.3. miRNA-92a

miR-92a is secreted from K562 leukemia cells and is shuttled to endothelial cells. Treatment with exosomal miR-92a was shown to regulates integrin α5 in HUVEC endothelial cell model. In this context miR-92a works as one of the effectors of the migration and the formation of angiogenetic structures [68]. 

## 6. Discussion

Overall, this review has pointed out the relevance of nuclear-cytoplasmic shuttling in CML, as well as it was defined in other cancers [69]. Various tumor suppressors, oncogenes and nucleic acids have been shown to shuttle from various cellular compartments in CML, promoting changes in the behaviors of the cells. Notably, shuttling of these proteins and mRNA was shown to be induced by BCR-ABL itself and to cooperate with BCR-ABL in the pathogenesis and/or maintenance of CML. Targeting shuttling mechanisms was conversely associated with the restoration of the normal cellular compartmentalization and functions of the proteins described in this review. Therefore, shuttling appears to be a perfect target in those scenarios where the inhibition of BCR-ABL is insufficient to promote eradication of the cancer: in the presence of BCR-ABL mutations that impair sensitivity to TKI, in the stem cell pool and in CML blast crisis/Ph+ ALL. Due to the tremendous impact of the inhibitor of Xpo1 in multiple myeloma [6], we may expect that targeting the nuclear-cytoplasmic machinery may be translated into Ph+ leukemias to provide new therapeutic chances.

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
