# Peer review of "Nuclear-cytoplasmic Shuttling in Chronic Myeloid Leukemia: Implications in Leukemia Maintenance and Therapy"

_cells, 2019, doi:10.3390/cells8101248_

Round 1

Reviewer 1 Report

The manuscript entitled “Nuclear-cytoplasmic shuttling in Chronic Myeloid Leukemia: implications in leukemia maintenance and therapy” is a review discussing some of the most relevant proteins or RNAs which, shuttling between nucleus and cytoplasm, affect CML maintenance and therapy.

The topic of the review is very interesting and the manuscript appears as a timely summary of all the findings currently available in the literature. However, different minor comments should be addressed.

Minor comments:

I would suggest to move the BCR-ABL1 paragraph (lines 51-66) in the Introduction, were it is more pertinent and will give a better overview of the oncoprotein’s nuclear-cytoplasmic shuttling in CML. Hence, the title of section 2 should change in “Shuttling tumor suppressors”. The sentence “and is involved in the negative regulation of the PI3K-Akt signaling [22]” in line 69 should be deleted because it is repeated immediately afterwards. I would suggest to better discuss the role of Foxo proteins in CML, referring to the cited bibliography (references 34-37). In the p53 paragraph a single reference (45) refers to p53 subcellular localization in CML. It is probably not enough, considering the topic of the review. I would suggest to better discuss the role of RBP in CML in the “BCR-ABL-dependent shuttle of RNA-binding proteins” paragraph, expanding what is reported in the cited references (52-54) (lines 115-116). I would also suggest to rethink the organization of the sub-paragraphs of section 4: 4. hnRNP (Heterogeneous nuclear ribonucleoproteins), including 4.1 hnRNP A1 and 4.2 Fus; 5. BCR/ABL-dependent shuttling of non-coding RNA, including 5.1 miRNA-126, 5.2 miRNA-210 and 5.3 miRNA-92a. When authors discuss BCR-ABL subcellular localization they should add the reference “Preyer, PlosOne, 2011”, which is one of the most recent on this topic. Authors’ names are missing in references 24 and 29. However, reference 29 should be removed (line 79), as it is the same of reference 30, without the authors’ name. Please write “nuclear-cytoplasmic shuttling” always in the same way. Please change “explained” with “explain” in line 78. Please substitute “RNA binding proteins” in line 115 with “RBPs”. Please substitute “phosphatidylinositol 3-kinase” with PI3K in line 125. Please check the English language in sentence in line 156-157.

Author Response

We thank reviewer’s comments that allow us to further improve our manuscript.

1) we move BCR-ABL paragraph in the introduction, as suggested.

2) we changed the title of the Shuttling tumor suppressors section, as suggested

3) The sentence “and is involved in the negative regulation of the PI3K-Akt signaling [22]” in line 69 has been deleted.

4) the role of FOXO has been better defined

5) we agree that p53 as a shuttling protein in CML did include only one reference. We have extended this part.

6)  we have better described  the role of RBP in CML in the “BCR-ABL-dependent shuttle of RNA-binding proteins” paragraph as suggested

7) we have included sub-paragraphs of sections

8) we agree with the suggestion to include the reference “Preyer, PlosOne, 2011”

9) references 24 and 29 have been corrected.

10) “nuclear-cytoplasmic shuttling” was written always in the same way.

11) “explained” with “explain” in line 78 was corrected

12) Please substitute “RNA binding proteins” in line 115 with “RBPs”. DONE

13) Please substitute “phosphatidylinositol 3-kinase” with PI3K in line 125: done.

14) Please check the English language in sentence in line 156-157: done

Reviewer 2 Report

The authors of this review provide a brief overview of how nuclear-cytoplasmic shuttling of proteins and/or RNA can affect the biology of leukemic transformation, with special emphasis on Chronic Myeloid Leukemia. They discuss data on some shuttling proteins and protein-RNA complexes important for leukemias, which show that aberrations of the normal shuttling of these proteins can be found in leukemic cells. The authors highlight the therapeutic potential of restoring this perturbed shuttling of proteins.

I have a few recommendations that I believe would improve the quality and readability of the review. Given in order of importance, these are:

1) Sections 4.3 to 4.6 on page 4 refer to extra-cellular shuttling of molecules (through exosomes), rather than intra-cellular shuttling between the nucleus and the cytoplasm. Therefore, they fall out of the subject of this review, which is "nuclear-cytoplasmic shuttling", at least as this is stated clearly in the title and abstract. I would recommend either to remove these sections altogether, or to change the title of the review accordingly. In the latter case, I would recommend to try to expand a bit more on these "extra-cellular" shuttling proteins, or otherwise keep them all under the same paragraph. The authors give only little information - essentially each section refers to one or two publications.

2) Two references should be corrected: Ref. 29 is essentially ref. 30 without the author names. Presumably some other reference should be inserted there, since it should be related to PTEN (section 2.2) and not to Foxo (section 2.3). Also, ref. 24 is given without the author names.

3) I would strongly recommend some language editing and, if possible, a reviewing of the manuscript by a native english speaker. Quite a few sentences are not clearly written and require a second or third reading in order to understand what the authors mean. An example of this is the sentence in lines 125-127. There are also several grammatical mistakes or even typos, so some more care should be taken in order to improve clarity.

Author Response

I wish to thank reviewer’s comments that allowed to improve our manuscript.

1) we agree with this comment. It is a major issue. We have attempted to better explain the reason of this chapted in the introduction of this section. However, if it still remains unclear, we may also consider removing it.

2) references have been corrected

3) English has been revised.